# Protective Encapsulation of a Bioactive Compound in Starch–Polyethylene Glycol-Modified Microparticles: Degradation Analysis with Enzymes

**DOI:** 10.3390/polym16142075

**Published:** 2024-07-20

**Authors:** Karen Sofia Valenzuela Villela, Karen Valeria Alvarado Araujo, Perla Elvia Garcia Casillas, Christian Chapa González

**Affiliations:** 1Grupo de Investigación Nanomedicina, Instituto de Ingeniería y Tecnología, Universidad Autónoma de Ciudad Juárez, Ciudad Juárez 32310, Mexico; karen.valenzuela@uacj.mx (K.S.V.V.); al168591@alumnos.uacj.mx (K.V.A.A.); 2Departamento de Física y Matemáticas, Instituto de Ingeniería y Tecnología, Universidad Autónoma de Ciudad Juárez, Ciudad Juárez 32310, Mexico; 3Centro de Investigación en Química Aplicada, Blvd. Enrique Reyna Hermosillo No. 140, Saltillo 25294, Mexico

**Keywords:** biomaterial, starch, enzyme, amylase, pepsin, drug delivery, polyethylene glycol, folic acid, microencapsulation, degradation

## Abstract

Starch is a promising polymer for creating novel microparticulate systems with superior biocompatibility and controlled drug delivery capabilities. In this study, we synthesized polyethylene glycol (PEG)-modified starch microparticles and encapsulated folic acid using a solvent-mediated acid-base precipitation method with magnetic stirring, which is a simple and effective method. To evaluate particle degradation, we simulated physiological conditions by employing an enzymatic degradation approach. Our results with FTIR and SEM confirmed the successful synthesis of starch–PEG microparticles encapsulating folic acid. The average size of starch microparticles encapsulating folic acid was 4.97 μm and increased to 6.01 μm upon modification with PEG. The microparticles were first exposed to amylase at pH 6.7 and pepsin at pH 1.5 at different incubation times at physiological temperature with shaking. Post-degradation analysis revealed changes in particle size and morphology, indicating effective enzymatic degradation. FTIR spectroscopy was used to assess the chemical composition before and after degradation. The initial FTIR spectra displayed characteristic peaks of starch, PEG, and folic acid, which showed decreased intensities after enzymatic degradation, suggesting alterations in chemical composition. These findings demonstrate the ongoing development of starch–PEG microparticles for controlled drug delivery and other biomedical applications and provide the basis for further exploration of PEG–starch as a versatile biomaterial for encapsulating bioactive compounds.

## 1. Introduction

Polymeric microparticles have attracted great interest in recent years due to their versatile performance in various biomedical applications [1,2,3,4,5]. These microparticles can encapsulate a wide range of molecules, providing protection from degradation and facilitating their controlled release [6,7]. By encapsulating therapeutic agents, whether active ingredients or bioactive compounds, these microparticles improve solubility, stability, and bioavailability, with the potential to improve efficacy and selectivity in treatments. In recent years, much progress has been made in the development of polymeric microparticles designed to release their specific payload in response to stimuli, such as changes in pH [8] or enzymatic activity [9] which can be exploited for cancer therapy.

Cancer is a complex multifactorial disease that affects millions of people around the world each year [10]. Our knowledge of many of the causes of cancer has led to progress towards new forms of treatment, diagnosis, and prevention. For example, recent studies suggest that folic acid (FA) supplementation, especially before and during pregnancy, might help prevent certain cancers like acute lymphoblastic leukemia (ALL) and acute myeloid leukemia (AML) by lowering the risk of chromosomal abnormalities [11]. In contrast, folate compounds act as cofactors in the one-carbon metabolism, and the central function in the feeding of rapidly dividing cancer cells makes it an attractive target in hematological malignancies [12]. Antifolates such as methotrexate take advantage of this dependence, but their limitations make more specific strategies necessary. Research is focused on enzymes regulated in cancer cells within the folate pathway to improve therapies. For this reason, the concept of folic acid in health and disease has recently been reconsidered [13], recommending a prudent approach to the supply that would be safer and more effective, for example, in Hispanic pregnant subjects of lower income and education [14].

Regardless, new ways of administering FA, and *N*-acyl-amino acid, are being sought because of the benefits it presents and the disastrous consequences caused by its deficiency. FA deficiency leads to anemia with impaired hemoglobin synthesis and cell maturation. Severe deficiency can cause reduced thymidylate synthesis and abnormal uracil incorporation into DNA, resulting in DNA strand breaks. FA, or vitamin B9, must be ingested through the diet or supplements. FA is obtained naturally by several foods; for example, legumes, eggs, citrus fruits, seeds and pecan nuts [15]. However, cooking can degrade folate in foods [16]. In addition, low pH can degrade FA [17], reducing absorption. Thus, polymeric carrier formulations that protect folic acid from the low pH gastric environment are highly sought after to enhance stability and absorption, increasing bioavailability. In general, the use of polymeric systems to encapsulate bioactive compounds aims to enhance its protection against degradation, improve solubility, and increase bioavailability, addressing key challenges in the development of effective nanostructures for drug delivery.

In recent decades, there have been significant advances in delivery systems, which has enabled more effective drug delivery [18,19,20,21,22,23,24]. To deliver active ingredient, genes [25,26], drugs [27], or bioactive compounds [28] more efficiently, nanoparticle [29], liposome, micelle [30], magnetic [31] or polymeric delivery systems [32,33] are being designed. In that sense, formulations with starch polymers can be a strategy to encapsulate the FA or any bioactive compound or drug and at the same time deliver it by the action of the enzyme amylase when it hydrolyzes the coating and releases the contents. Enzymes can be employed as a trigger in the design of release systems due to their unique characteristics such as substrate specificity and high selectivity. Since enzymes are related to almost all biological and metabolic processes, as they can be exploited to achieve drug release through biocatalytic action, they can be used as a trigger in the design of release systems.

Starch is a polymer with a wide range of potential biomedical applications including tissue engineering and drug delivery [34,35,36,37]. Starch is a storage polymer composed of two types of glucose molecules: amylose, which is a linear molecule that represents about 20–25% of starch and is coiled into a helix, and amylopectin, a branched molecule that makes up the remaining starch and has a more complex structure. The glycosidic bonds that bind the successive glucose residues in the amylopectin chains are (α1→4), and the branching points (which occur every 24 to 30 residues) are bonds (α1→6) [38]. The versatility of starch makes it an attractive polymer for the development of encapsulation and release systems for pharmaceuticals [39,40,41,42,43,44], in this case for folic acid. Likewise, poly(ethylene glycol) (PEG) exhibits numerous advantages for use in biomedical applications, such as surface modifications of nanoparticles [45,46], hydrogels [47,48], and formulations of particles carrying active ingredients [49], primarily due to its excellent biodegradability, biocompatibility, and history of medical use. PEG can enhance drug delivery by slowing down drug clearance, shielding protein therapeutics from undesirable immunogenicity, and creating stealth drug carriers with prolonged circulation time and decreased recognition and clearance by mononuclear phagocytic cells.

Thus, PEG–starch conjugations have been utilized in several key ways. Research has demonstrated that the mixture of starch and PEG at different molecular weights can influence the starch chain conformation and gelatinization properties [50]. Other studies have investigated moisture barrier properties in optimized PEG–starch composites for packaging applications [51]. Additionally, dual-functional hemostatic sponges based on PEG–starch for controlling uncontrolled hemorrhage have been developed [52]. In contrast, this work focuses on the encapsulation of bioactive compounds in PEG-modified starch microparticles, specifically evaluating enzymatic degradation in simulated physiological conditions. In this study, we aim to develop a starch–PEG system encapsulating FA (FA-S-PEG) to investigate its stability under gastric conditions in the presence of pepsin, while also comparing the effects of amylase. Given that drug absorption can be influenced by various physicochemical variables, such as pH-dependent solubility and stability, enzymatic degradation, and other physiological agents that operate through transport mechanisms in specific regions of the gastrointestinal system, we examined the behavior of starch–PEG microparticles under simulated gastric conditions. This involved monitoring particle size and evaluating functional groups over different time intervals, correlating with the residence time in each compartment of the gastrointestinal system. Our objective was to determine whether the formulation protects FA from degradation by gastric pH, with PEG providing protection and amylase contributing to controlled release.

## 2. Materials and Methods

### 2.1. Materials

Starch from rice (CAS 9005-25-8; S7260), poly(ethylene glycol) (PEG, BioUltra 2000, CAS 25322-68-3; 84,797), and folic acid (CAS 59-30-3, F7876) were purchased from Sigma-Aldrich (St. Louis, MO, USA), while acetone (Golden Bell, México City, México, ACS), hydrochloric acid (J.T. Baker, Phillipsburg, NJ, USA, ACS), and ammonium hydroxide (Fermont, Monterrey, Nuevo León, México, ACS) were purchased separately. All reagents were used as received without any further purification. The enzymes α-amylase (Enziquim, Ciudad de México, México, E.C. 3.2.1.1., 32,000–36,000 BAU/g) and pepsin (Nutricost, Vineyard, UTAH, USA, E.C. 3.4.23.1., 1:10,000), both food grade, were obtained commercially.

### 2.2. Preparation of Microparticules

The method of processing the microparticles was followed according to the procedure indicated elsewhere [53]. To initiate the overlay of the powdered folic acid with the starch, 0.2 g of folic acid and 1.0 g of rice starch were weighed, with the aid of a balance. In a graduated cylinder, 50 mL of acetone was added and later, the 0.2 g of folic acid and 1.0 g of starch were added, with the help of a previously calibrated potentiometer. The pH of the sample was measured and 5% hydrochloric acid was poured drop by drop until the sample a pH of 2.5. A magnetic stir bar was added to the beaker with the sample, placing the beaker on a magnetic stirring plate, covering the beaker with aluminum so as not to contaminate the sample. Then, 100 mL of distilled water was added to a burette and placed in the sample to be emptied drop by drop during the stirring process for 20 min, so that the sample reached a pH of approximately 6.0. The obtained mixture was emptied in conical centrifuge tubes to pass directly to the centrifuge for 10 min at 3000 rpm. Subsequently, 3 washes with ethanol were performed, each wash for 5 min at 3000 rpm, in order to obtain the sample. Then, the sample was poured into a covered crucible and placed in the drying oven at 30 °C for 24 h. 

The modification of folic acid–starch with polyethylene glycol was started by weighing 0.5 g of PEG 2000 and 0.05 g of the previously dried folic acid–starch microparticles and this was pulverized with the aid of a mortar. In a test tube, 30 mL of distilled water was emptied and poured into a beaker. The beaker was placed on a plate at a temperature of 40 °C, and then the 0.05 g of starch–folic acid was added, before placing this in a magnetic agitator device for 5 min. Subsequently, the 0.5 g of PEG 2000 was added, continuing magnetic agitation for 5 more minutes. Once the time had elapsed, the pH was modified to 9 with the help of 5% ammonium hydroxide previously prepared (5–6 drops approx.) and magnetic agitation was continued for 20 min. After a period of time, the sample was removed from the agitator and was poured into falcon tubes to be centrifuged for 5 min at 3000 rpm, washed 3 times with distilled water, and then centrifuged for the same time and rpm, until it reached a pH of 7. Finally, after reaching the pH, the sample was placed in a crucible and it was left to dry at 50 degrees Celsius in the oven for 24 h. Once the 24 h had passed, the sample was removed from the oven and placed in a mortar to be pulverized; then, the sample was stored in a container. This methodology was carried out approximately 24 times in order to obtain the following results.

### 2.3. Characterization

Once the starch–folic acid microparticles were obtained, the characterization proceeded. With the help of Fourier transform infrared spectroscopy (Thermo Scientific, Shanghai, China, iS50 ATR), a pinch of the microparticles was placed in the equipment in order to obtain the bands with their functional groups. The infrared spectra were recorded with a resolution of 8 cm^−1^, and the scan range was set from 4000 to 600 cm^−1^. The results are presented as the average of 32 scans. This procedure was repeated with FA-S-PEG microparticles. After this, a sample of the FA-S microstructures was placed in the Scanning electron microscopy equipment, using a Hitachi equipment (SU5000) operating at 15 kV. The samples were directly detached on a doubled-face carbon conductive tape before SEM observation to determine their morphology and average size by measuring the size of 100 individual microparticles using ImageJ software (version 1.54 g). This process was repeated with the FA-S-PEG microstructures. 

### 2.4. Enzimatic Degradation of FA-S-PEG in Physiological Simulated Conditions

To prepare the simulated gastric acid, 7 mL of HCl and 2 g of NaCl were added to a beaker. The pH was measured to confirm that a pH of about 1.2 was obtained, and the enzyme pepsin was added and placed under magnetic stirring for 20 min. To conduct the degradation, different times were set; in the case of gastric acid simulated with pepsin, the degradation was set at 30, 60, and 180 min, and in the case of amylase pH 6.7, the times were set at 5, 15, and 180 min. The different durations for enzyme treatments were chosen to reflect their specific roles in simulated physiological conditions, providing relevant time points for comprehensive analysis [54]. Approximately 5 L of ionized water was added to the water bath equipment and the temperature was adjusted to 37 °C, simulating body temperature. Then, in 6 separate Eppendorf tubes, 0.025 g of the FA-S-PEG microparticles was added, and to 3 tubes, 1.5 mL of the simulated gastric acid was added. To the remaining 3 tubes, 1.5 mL of amylase was added. Subsequently, the tubes were capped and placed in the water bath, with each one assigned to the time already established and mentioned previously; once the time of each one had passed, they were removed from the water bath, centrifuged at 3000 rpm for 5 min, and 3 washes were conducted for each tube with distilled water, centrifuging each wash at 3000 rpm for 5 min. Once the samples were dried, they were characterized by FTIR and SEM to observe their functional groups and determine if there was a significant degradation at the different times already established.

### 2.5. Statistical Analysis

A statistical analysis using one-way Analysis of Variance (ANOVA) followed by Tukey’s Honestly Significant Difference (HSD) post hoc test was conducted. ANOVA was used to determine if there were significant differences in the average size of the microparticles over time for each enzyme treatment. Tukey’s HSD test was applied post hoc to identify specific time points with statistically significant differences in particle size.

## 3. Results and Discussion

The synthesis of starch microparticles encapsulating folic acid yielded particles with an average size of 4.97 ± 1.19 μm. Following the PEG modification, the average size increased to 6.01 ± 1.52 μm, indicating successful coating with PEG. The SEM images in Figure 1 provide visual confirmation of these observations. The increase in particle size post-PEG modification suggests that the PEG coating was successfully applied to the starch microparticles. This is further supported by the distinct morphology changes observed in the SEM images, where the FA-S-PEG particles appear larger compared to the AF-S particles. The histograms of size distribution (Figure 1c) also show a shift towards larger particle sizes, which is consistent with the PEG coating adding an additional layer to the microparticles. 

The FTIR spectroscopy results provide compelling evidence for the successful encapsulation of FA within starch microparticles and the subsequent modification with PEG. In Figure 2a, the FTIR spectra of folic acid-encapsulated starch microparticles (FA-S) are presented. The spectrum exhibits prominent absorption bands at 1033 cm−^1^, which correspond to the C-O bond characteristic of starch. Additionally, the band at 990 cm−^1^ is attributed to α-1,4 glycosidic linkages and C-O-C stretching vibrations, indicative of the structural presence of starch. The inclusion of folic acid is evidenced by absorption bands in the range of 1690–1500 cm−^1^, associated with the carboxylic acid groups, and a band at 1250 cm−^1^, attributed to the C-C bond. These spectral features confirm the successful encapsulation of folic acid within the starch matrix, as both starch and folic acid characteristic bands are discernible. Figure 2b displays the FTIR spectra of PEG-modified starch microparticles encapsulating folic acid (FA-S-PEG). In addition to the absorption bands observed for starch and folic acid, new bands appear at 1098 cm−^1^, corresponding to the C-O-C bond, at 971 cm−^1^, and at 2859 cm−^1^, corresponding to the stretching vibrations of CH. These bands are characteristic of PEG, indicating its successful coating on the microparticles. The spectra thus confirm the presence of starch, folic acid, and PEG, demonstrating the successful synthesis and modification of the microparticles.

The FTIR spectroscopy results depicted in Figure 3 provide data on the degradation behavior of AF-A-PEG microparticles in the presence of the enzymes under simulated physiological conditions. Figure 3a presents the FTIR spectra of AF-A-PEG microparticles subjected to degradation at various time intervals: 5, 15, and 180 min. Key absorption bands corresponding to the functional groups of folic acid, starch, and PEG are observed. For folic acid, the N-H and O-H stretching vibrations at 3275 cm−^1^ become more prominent after 180 min of degradation, indicating the persistence and prominence of the carboxyl group under prolonged enzymatic exposure. The starch component shows characteristic bands and for PEG, methylene group stretches (CH, CH_2_, and CH_3_) are noted at 2869 cm−^1^, along with O-H stretching attributable to hydroxyl groups. In Figure 3b, the FTIR spectra for AF-A-PEG microparticles subjected to 180 min of degradation with either amylase (D180A) or pepsin (D180P) are compared. The characteristic bands for each molecule are consistently present. However, notable differences emerge between the two enzymatic treatments. The spectra show less pronounced bands for amylase-degraded samples compared to those treated with pepsin. This observation can be attributed to amylase’s specific substrate: starch. Amylase catalyzes the hydrolysis of glycosidic bonds in starch, breaking it down into smaller sugars such as maltose and glucose, which are more readily absorbed and utilized by the body. This enzymatic action results in the diminished visibility of starch-related bands in the FTIR spectra. The presence of less intense bands in the amylase-degraded samples is indicative of significant structural decomposition. When starch is degraded by amylase, its chemical bonds are cleaved, leading to the loss or modification of original functional groups. Consequently, the characteristic FTIR bands associated with these groups may weaken, shift, or even disappear. In contrast, pepsin, which primarily targets protein substrates, does not significantly affect the starch component, allowing the characteristic bands to remain more noticeable. The more pronounced degradation observed with amylase highlights its efficacy in breaking down starch-based carriers, which has significant implications for the design and optimization of enzyme-responsive drug delivery systems. The resilience of folic acid bands and the presence of PEG-related bands across different degradation conditions further confirm the structural integrity of these components within the microparticles, suggesting that they can effectively protect and release the encapsulated folic acid under gastric enzymatic conditions.

The morphological changes in FA-S-PEG microparticles subjected to enzymatic degradation reveal a progressive alteration over time (Figure 4). Initially, after 5 min of exposure to amylase (Figure 4a), the microparticles exhibit a relatively smooth and intact surface, indicating minimal enzymatic interaction. By 15 min, noticeable surface erosion begins, characterized by the appearance of irregularities and indentations on the particle surfaces. This suggests that the amylase enzyme has started to hydrolyze the starch component within the microparticles. At 180 min, the degradation is significantly more pronounced. The microparticles display extensive surface roughness and fragmentation, with some particles appearing to have disintegrated entirely. The advanced stage of enzymatic activity by amylase indicates substantial hydrolytic breakdown, resulting in a highly porous and disrupted morphology. These observations align with the expected action of amylase, which specifically targets glycosidic bonds in polysaccharides, leading to their progressive disintegration. 

In contrast, the degradation of FA-S-PEG microparticles by pepsin (Figure 4b) demonstrates a different morphological evolution. After 30 min, the particles show initial signs of surface pitting and minor erosions. The action of pepsin, a proteolytic enzyme, initially appears less aggressive compared to amylase, reflecting its substrate specificity for peptide bonds rather than polysaccharides. At 60 min, the microparticles exhibit more extensive pitting and they are stuck to each other, suggesting an acceleration in activity, attributed to erosion or degradation of the surface due to low pH. The surface morphology becomes increasingly irregular. By 180 min, the microparticles display significant structural compromise, with large portions of the surface eroded and agglomerated, so much so that it is difficult to notice the initial morphology and edges of the microparticles. Comparing the morphological changes induced by amylase and pepsin, it is evident that the enzymatic degradation pathways differ significantly due to the enzymes’ substrate specificities. Amylase induces rapid and extensive surface erosion, resulting in a highly porous and fragmented morphology within a shorter timeframe. This suggests that the starch component in FA-S-PEG microparticles is particularly susceptible to amylase action.

Figure 5 consolidates the time-dependent changes in microparticle size, derived from the individual data points presented in Figure 4, and it examines the size reduction in PEG-modified starch microparticles encapsulating folic acid when exposed to the amylase and pepsin enzymes under simulated physiological conditions. The preparation involved encapsulating folic acid in PEG-coated starch microparticles and incubating them with either amylase at pH 5 or pepsin at pH 1.2, reflecting the enzyme activity in the mouth and stomach, respectively. The size of the microparticles was measured over time using SEM, with 100 particles analyzed at each time point. The results indicate that a PEG coating offers substantial protection to the starch microparticles in highly acidic environments. This protection has an effect in preserving the structural integrity of the microparticles when exposed to the harsh gastric conditions simulated by the pepsin at pH 1.2. Moreover, the data reveal that amylase leads to a more pronounced reduction in particle size. This finding can be leveraged to control the release of encapsulated substances. Since amylase operates at a less acidic pH and directly targets starch, it effectively reduces the size of the microparticles, potentially accelerating the release of the encapsulated folic acid. This enzymatic activity suggests that in environments where amylase is active, such as in the mouth and small intestine, controlled degradation of the microparticles can be achieved.

The size reduction in microparticles over time under the action of amylase and pepsin enzymes was observed. However, statistical analysis shows that these reductions are not significantly different across the enzyme treatments. Based on the data, an ANOVA test followed by Tukey’s HSD test to assess the differences in microparticle size over time in the presence of amylase and pepsin enzymes were conducted. With a *p*-value greater than 0.05, the ANOVA test suggests that there is no statistically significant difference in the average size of the microparticles when comparing the groups treated with different enzymes. According to Tukey’s HSD test, there are no statistically significant differences between any pairs of groups at the 0.05 significance level. This indicates that while both enzymes contribute to the degradation of microparticles, their effects are not significantly distinct from each other or from the untreated control within the time frame studied. 

In previous studies, researchers observed that PEG is distributed in a starch matrix with PLA and obtained higher degradation rates under basic conditions and concluded that the main mechanism of polymer biodegradation was hydrolysis by enzymes [55]. Our results also indicated that enzymatic degradation affects the structural integrity of the microparticles, which reinforces the involvement of enzymatic activity in polymer degradation pursuing slow release of compounds from biocompatible and biodegradable matrices, as indicated previously [56]. Nevertheless, since pepsin is a proteolytic enzyme, it primarily acts on proteins and shows limited activity on starch-based materials. This specificity accounts for the minimal size reduction observed in the presence of pepsin. The graph supports this by showing only slight changes in particle size over time when exposed to pepsin. Although starch is a natural substrate for amylase, the modification within PEG results in only moderate enzymatic activity. The PEG coating likely impedes the amylase from fully accessing the starch, thus slowing down the degradation process. This suggests that while amylase can degrade starch, the presence of PEG modifies the interaction, reducing the overall rate of size reduction. These findings show the potential application of PEG–starch microparticles in controlled delivery systems. The ability to successfully encapsulate folic acid and to observe the change in the average size of the microparticles as a result of its degradation by enzymatic action highlights the effectiveness of microparticles made with these polymers to protect bioactive compounds. The microparticles allow the compound to be protected from the environment of the gastrointestinal tract, making these microparticles a promising candidate for targeted delivery applications in biomedicine. 

## 4. Conclusions

The findings of this study highlight the importance of encapsulation in PEG-coated starch microparticles of molecules with pharmacological interest to protect them from environments of the gastrointestinal tract. Fourier transform infrared spectroscopy (FTIR) and scanning electron microscopy (SEM) demonstrated the successful synthesis of these microparticles. Furthermore, this study showed that amylase caused a slightly greater reduction in particle size over 180 min compared to pepsin, which was attributed to the fact that starch is a substrate of amylase. Despite being PEG coated, the starch microparticles still showed a remarkable size reduction when exposed to amylase, confirming that starch is degraded even with the PEG barrier. On the other hand, the change in morphology with pepsin can be attributed more to the low pH environment than to the enzymatic activity of pepsin since pepsin mainly targets proteins. Furthermore, FTIR spectra corroborated the findings, reinforcing the debate on the stability and degradation of the folic acid–starch–PEG system under simulated gastric conditions with pepsin and amylase. While our study demonstrates the potential of PEG–starch microparticles for controlled drug delivery, further research is needed to optimize the encapsulation efficiency and release kinetics for various bioactive compounds. Additionally, in vivo studies are required to validate the biocompatibility and therapeutic efficacy of these microparticles in real physiological conditions. Future experiments should also explore the long-term stability of the encapsulated agents and the scalability of the synthesis process to ensure practical applications in biomedical fields. Nevertheless, it serves as a foundational basis for exploring PEG-modified starch microparticles as biomaterials for encapsulating bioactive compounds, genes, drugs, and active ingredients, thereby potentially providing protection in environments where these molecules may undergo degradation.

## Figures and Tables

**Figure 1 polymers-16-02075-f001:**
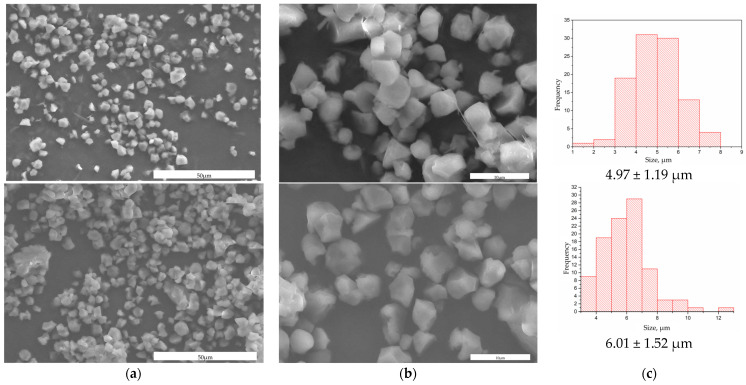
(**a**) The upper panel shows micrographs of starch microparticles encapsulating folic acid (AF-S), while the lower panel shows PEG-modified starch microparticles encapsulating folic acid (AF-S-PEG), scale bar 50 μm; (**b**) the upper panel shows AF-S, scale bar 10 μm; (**c**) histograms of size distribution, *n* = 100.

**Figure 2 polymers-16-02075-f002:**
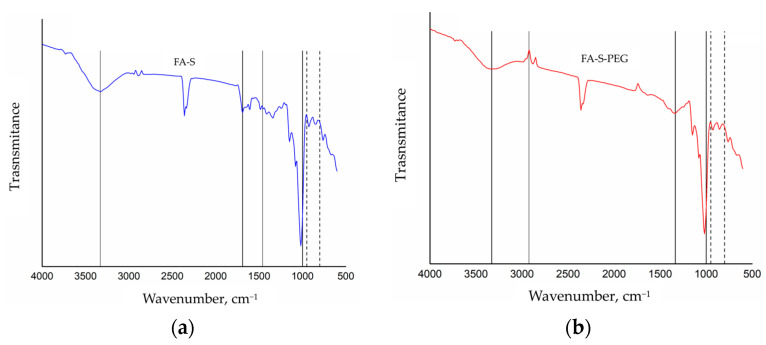
(**a**) FTIR spectrum of FA-S material; (**b**) FTIR spectrum of FA-S-PEG material.

**Figure 3 polymers-16-02075-f003:**
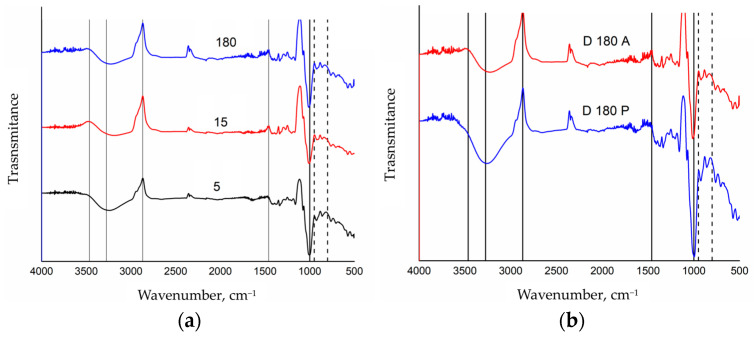
(**a**) FTIR spectra of AF A PEG microparticles subjected to degradation in the presence of amylase enzyme under simulated physiological conditions for 5, 15, and 180 min as indicated in each spectrum; (**b**) FTIR spectra of AF A PEG microparticles subjected to degradation for 180 min in the presence of amylase enzyme D180A and pepsin enzyme D180P.

**Figure 4 polymers-16-02075-f004:**
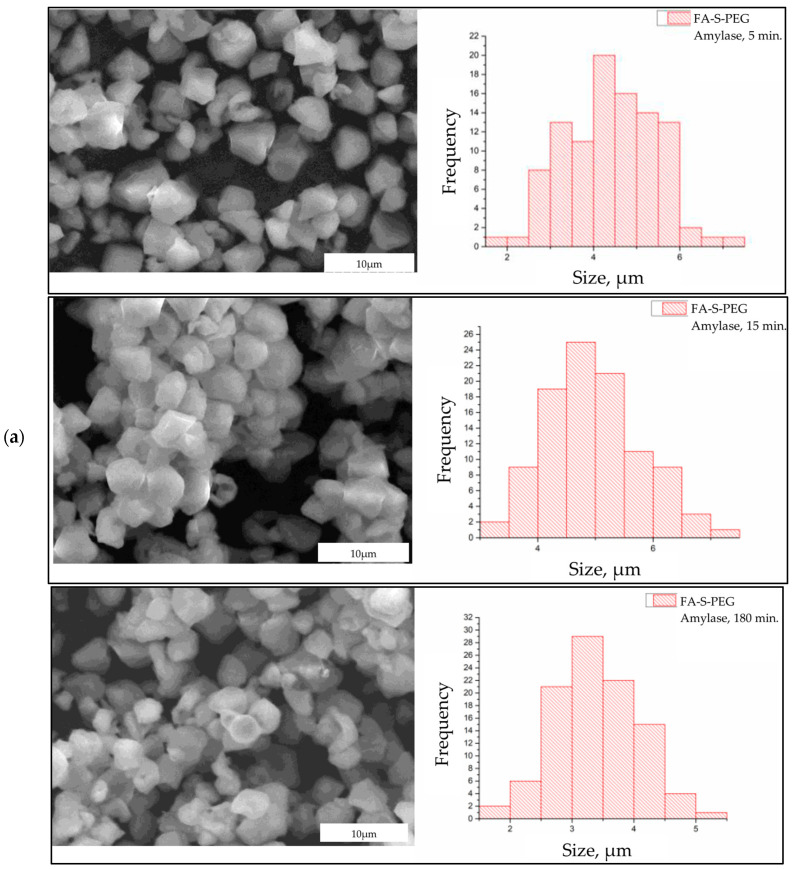
(**a**) Micrographs of FA-S-PEG microparticles subjected to degradation in the presence of amylase enzyme for 5, 15, and 180 min; (**b**) micrographs of FA-S-PEG microparticles subjected to degradation for 30, 60, and 180 min in the presence of pepsin.

**Figure 5 polymers-16-02075-f005:**
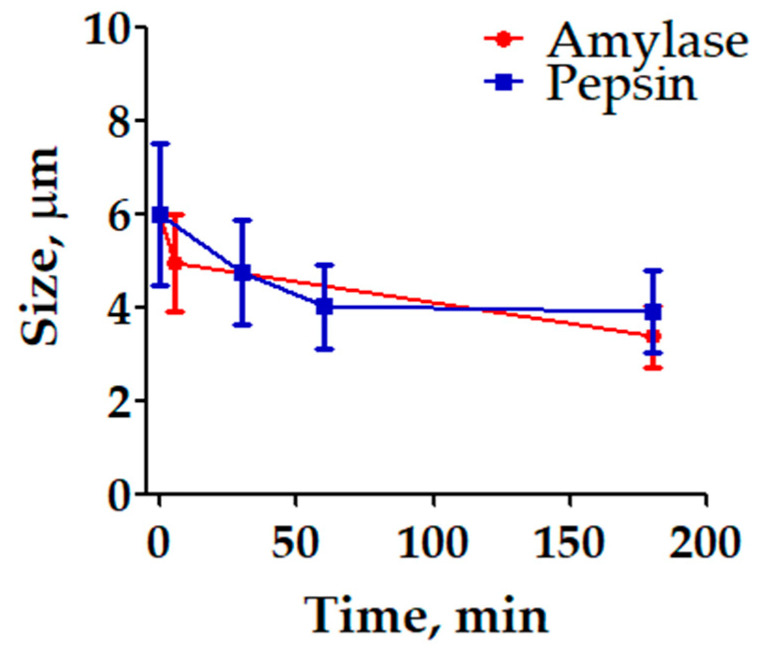
Size of FA-S-PEG microparticles encapsulating folic acid over time when incubated with amylase (pH 5) and pepsin (pH 1.2). Error bars represent standard deviation of 100 particle measurements.

## Data Availability

The raw data supporting the conclusions of this article will be made available by the authors on request.

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
