# Peer review of "Protective Encapsulation of a Bioactive Compound in Starch–Polyethylene Glycol-Modified Microparticles: Degradation Analysis with Enzymes"

_polymers, 2024, doi:10.3390/polym16142075_

Round 1

Reviewer 1 Report

Comments and Suggestions for Authors

The work entitled “Protective Encapsulation of a Bioactive Compound in Starch-Polyethylene Glycol-Modified Microparticles: Degradation Analysis with Enzymes” reports on the synthesis of polyethylene glycol (PEG)-modified starch microparticles encapsulating folic acid using a solvent-mediated acid-base precipitation method, following the mapping of its enzymatic degradation in simulated physiological conditions.

Data confirmed the encapsulation of the folic acid and the ability of the enzymes to degrade the microparticles, showing their ability for potential uses in biomedicine, with a somewhat controlled release of the encased bioactive agent.

The work is very well put together, and organized. The subject of investigation is also clear and of interest. However, prior to publication I recommend the authors to implement some alterations:

-          The novelty of the research is not clear. How do these microparticles differ from other PEG/starch-based conjugations?

-          Why not use a model drug instead of folic acid to access the ability to encapsulate and release the entrapped agent?

-          There is no information about the materials used. Please add a section for materials.

-          The methodology for preparing the particles is very detailed but there are many details missing from the characterization approaches. For instance, no parameters were identified for any of the techniques used.

-          The presentation of the data is ok, even though in the enzyme action statistical analysis is missing. However, much of the discussion is not supported by the literature nor criticized by it.

-          Also, what do these results mean in the context of the potential application.

-          The weaknesses of the study must also be identified in the conclusions.

Comments on the Quality of English Language

English writing must be improved. There are some grammar mistakes along the manuscript and some senteces that require attention for clearer comprehension.

Author Response

Dear Reviewer 1,

Thank you for your thorough review and valuable feedback on our manuscript. We have carefully addressed all your comments and incorporated the necessary revisions to improve the quality of our work. We believe that these changes have significantly strengthened the manuscript, and we hope it now meets the high standards of the journal. We sincerely appreciate your time and effort in reviewing our paper and look forward to your positive response.

---***---

Reviewer 1

The work entitled “Protective Encapsulation of a Bioactive Compound in Starch-Polyethylene Glycol-Modified Microparticles: Degradation Analysis with Enzymes” reports on the synthesis of polyethylene glycol (PEG)-modified starch microparticles encapsulating folic acid using a solvent-mediated acid-base precipitation method, following the mapping of its enzymatic degradation in simulated physiological conditions.

Data confirmed the encapsulation of the folic acid and the ability of the enzymes to degrade the microparticles, showing their ability for potential uses in biomedicine, with a somewhat controlled release of the encased bioactive agent.

The work is very well put together, and organized. The subject of investigation is also clear and of interest. However, prior to publication I recommend the authors to implement some alterations:

- The novelty of the research is not clear. How do these microparticles differ from other PEG/starch-based conjugations?

Response: Thank you very much for your question. To highlight the novelty of our research, we have reviewed recent works on PEG/starch-based conjugations and highlighted their various applications in different contexts. We have added the following text to the manuscript discussing how our microparticles differ from other formulations:

Manuscript

Revised version

…and clearance by mononuclear phagocytic cells.

     In this study, we aim to develop…

… and clearance by mononuclear phagocytic cells.

Thus, PEG/starch conjugations have been utilized in several key ways. Research has demonstrated that the mixture of starch and PEG at different molecular weights can influence the starch chain conformation and gelatinization properties [50]. Other studies have investigated moisture barrier properties in optimized PEG/starch composites for packaging applications [51]. Additionally, dual-functional hemostatic sponges based on PEG/starch for controlling uncontrolled hemorrhage have been developed [52] In contrast, this work focuses on the encapsulation of bioactive compound in PEG-modified starch microparticles, specifically evaluating enzymatic degradation in simulated physiological conditions. In this study, we aim to develop…

References added:

50.       Kim, C.H.; Kim, D.W.; Cho, K.Y. The Influence of PEG Molecular Weight on the Structural Changes of Corn Starch in a Starch/PEG Blend. Polymer Bulletin 2009, 63, 91–99, doi:10.1007/S00289-009-0065-8/

51.       Saputri, D.G.; Khairuddin; Nurhayati, N.D.; Pham, T. Optimization Properties of Environmentally Friendly Paper Coating Based Starch-Polyethylene Glycol (PEG) Mixture. J Phys Conf Ser 2017, 909, 012029, doi:10.1088/1742-6596/909/1/012029.

52.       Yang, X.; Liu, W.; Shi, Y.; Xi, G.; Wang, M.; Liang, B.; Feng, Y.; Ren, X.; Shi, C. Peptide-Immobilized Starch/PEG Sponge with Rapid Shape Recovery and Dual-Function for Both Uncontrolled and Noncompressible Hemorrhage. Acta Biomater 2019, 99, 220–235, doi:10.1016/J.ACTBIO.2019.08.039.

- Why not use a model drug instead of folic acid to access the ability to encapsulate and release the entrapped agent?

Response: Thank you for your question. We are interested in folic acid because one of the research lines increasingly used in polymer science is the encapsulation of bioactive compounds. While these compounds promote improvements in human health, they are often improperly absorbed. Thus, polymeric systems improve several characteristics, such as protection against degradation, solubility, stability, and bioavailability, among others. However, the development of nanostructures faces many challenges, from choosing the best method to obtain them to identifying the type of nanomaterial ideal for a bioactive compound of interest.

We have added the following text for clarification:

Manuscript

Revised version

… polymeric carrier formulations that protect folic acid from the low pH gastric environment are highly sought after to enhance stability and absorption, increasing bioavailability.

… polymeric carrier formulations that protect folic acid from the low pH gastric environment are highly sought after to enhance stability and absorption, increasing bioavailability. In general, the use of polymeric systems to encapsulate bioactive compounds aims to enhance its protection against degradation, improve solubility, and increase bioavailability, addressing key challenges in the development of effective nanostructures for drug delivery.

- There is no information about the materials used. Please add a section for materials.

Response: Thank you for your suggestion. We have added this information as the first subsection of Section 2. Materials and Methods.

Manuscript

Revised version

No Materials section

Materials

Starch from rice (CAS 9005-25-8; S7260), poly(ethylene glycol) (PEG, BioUltra 2,000, CAS 25322-68-3; 84797), folic acid (CAS 59-30-3, F7876) were purchased from Sigma-Aldrich, while acetone (Golden Bell, ACS), hydrochloric acid (J.T. Baker, ACS) and ammonium hydroxide (Fermont, ACS) were purchased separately. All reagents were used as received without any further purification.

- The methodology for preparing the particles is very detailed but there are many details missing from the characterization approaches. For instance, no parameters were identified for any of the techniques used.

Response: Thank you for your observation. We have revised the information on the methods to detail the characterizations employed.

Manuscript

Revised version

Characterization

Once the starch-folic acid microparticles were obtained, the characterization proceded. With the help of Fourier Transform Infrared Spectroscopy a pinch of the microparticles was placed in the equipment in order to obtain the bands with their functional groups. This procedure was repeated with FA-S-PEG microparticles. After this, a sample FA-S microstructures were placed in the Scanning Electron Microscopy equipment to determine their morphology and average size by measuring the size of 100 individual microparticles using ImageJ software. This process was repeated with the FA-S-PEG microstructures.

Characterization

Once the starch-folic acid microparticles were obtained, the characterization proceded. With the help of Fourier Transform Infrared Spectroscopy (Thermo Scientific, iS50 ATR), a pinch of the microparticles was placed in the equipment in order to obtain the bands with their functional groups. The infrared spectra were recorded with a resolution of 8 cm−1, and the scan range was set from 4000 to 600 cm−1. The results are presented as the average of 32 scans. This procedure was repeated with FA-S-PEG microparticles. After this, a sample FA-S microstructures were placed in the Scanning Electron Microscopy equipment, using a Hitachi equipment (SU5000) operating at 15 kV. The samples were directly detached on a doubled-face carbon conductive tape before SEM observation to determine their morphology and average size by measuring the size of 100 individual microparticles using ImageJ software. This process was repeated with the FA-S-PEG microstructures.

- The presentation of the data is ok, even though in the enzyme action statistical analysis is missing. However, much of the discussion is not supported by the literature nor criticized by it.

Response: Thank you very much for your comment. We highlight below the text added in the revised version of the manuscript concerning the enzyme action statistical analysis

Manuscript

Revised Version

No analysis

Statistical Analysis

A statistical analysis using one-way Analysis of Variance (ANOVA) followed by Tukey's Honestly Significant Difference (HSD) post-hoc test was conducted. ANOVA was used to determine if there were significant differences in the average size of the microparticles over time for each enzyme treatment. Tukey's HSD test was applied post-hoc to identify specific time points with statistically significant differences in particle size.

In Results and Discussion section:

The size reduction of microparticles over time under the action of amylase and pepsin enzymes was observed. However, statistical analysis shows that these reductions are not significantly different across the enzyme treatments. Based on the data, an ANOVA test followed by Tukey's HSD test to assess the differences in microparticle size over time in the presence of amylase and pepsin enzymes were conducted. With a p-value greater than 0.05, the ANOVA test suggests that there is no statistically significant difference in the average size of the microparticles when comparing the groups treated with different enzymes. According to the Tukey's HSD test, there are no statistically significant differences between any pairs of groups at the 0.05 significance level. This indicates that while both enzymes contribute to the degradation of microparticles, their effects are not significantly distinct from each other or from the untreated control within the time frame studied.

No discussion supported by the literature

In previous studies, researchers observed that PEG is distributed in a starch matrix with PLA and obtained higher degradation rates under basic conditions and concluded that the main mechanism of polymer biodegradation was hydrolysis by enzymes [55]. Our results also indicated that enzymatic degradation affects the structural integrity of the microparticles, which reinforces the involvement of enzymatic activity in polymer degradation pursuing slow release of compounds from biocompatible and biodegradable matrices as indicated previously [56]. Nevertheless,

References added:

55.       Momeni, S.; Ghomi, E.R.; Shakiba, M.; Shafiei-Navid, S.; Abdouss, M.; Bigham, A.; Khosravi, F.; Ahmadi, Z.; Faraji, M.; Abdouss, H.; et al. The Effect of Poly (Ethylene Glycol) Emulation on the Degradation of PLA/Starch Composites. Polymers 2021, Vol. 13, Page 1019 2021, 13, 1019, doi:10.3390/POLYM13071019.

56.       Akakuru, O.U.; Louis, H.; Uwaoma, R.; Elemike, E.E.; Akakuru, O.C. Novel Highly-Swellable and PH-Responsive Slow Release Formulations of Clotrimazole with Chitosan-g-PEG/Starch Microparticles. React Funct Polym 2019, 135, 32–43, doi:10.1016/J.REACTFUNCTPOLYM.2018.12.004.

- Also, what do these results mean in the context of the potential application.

Response: Thank you for your question. These results demonstrate the potential of PEG-starch microparticles for use in controlled delivery systems. The successful encapsulation and subsequent studies of enzymatic degradation indicate their capability to protect bioactive compounds like folic acid highlighting their suitability for biomedical applications, particularly in targeted delivery.

The following is the text we added in the revised version:

Manuscript

Revised version

… the presence of PEG modifies the interaction, reducing the overall rate of size reduction.

… the presence of PEG modifies the interaction, reducing the overall rate of size reduction. These findings show the potential application of PEG-starch microparticles in controlled delivery systems. The ability to successfully encapsulate folic acid and to observe the change in the average size of the microparticles as a result of its degradation by enzymatic action highlights the effectiveness of microparticles made with these polymers to protect bioactive compounds. The microparticles allow the compound to be protected from the environment of the gastrointestinal tract, making these microparticles a promising candidate for targeted delivery applications in biomedicine.

- The weaknesses of the study must also be identified in the conclusions.

Response: Thank you for your suggestion. The conclusions section now includes a discussion of the limitations of our study.

Manuscript

Revised version

… with pepsin and amylase. While this study provides valuable information, its limitations include the focus on a limited range of conditions and variability in particle size, as indicated by large standard deviations, but general trends were evident in mean size reductions. Nevertheless…

…with pepsin and amylase. While our study demonstrates the potential of PEG-starch microparticles for controlled drug delivery, further research is needed to optimize the encapsulation efficiency and release kinetics for various bioactive compounds. Additionally, in vivo studies are required to validate the biocompatibility and therapeutic efficacy of these microparticles in real physiological conditions. Future experiments should also explore the long-term stability of the encapsulated agents and the scalability of the synthesis process to ensure practical applications in biomedical fields. Nevertheless…

Reviewer 2 Report

Comments and Suggestions for Authors

Manuscript polymers-3108684 provides new information about the importance of encapsulation in polyethylene glycol-coated starch microparticles of molecules with pharmacological interest to protect them from environments of the gastrointestinal tract. There are a number of shortcomings in the work.

It seems to me inappropriate to include "enzymes" in the title. There is insufficient data on both the most used enzymes (or enzyme preparations) and the mechanism of the studied enzymatic processes.

In the introduction, the authors immediately tell about cancer, but the paper itself does not even contain data on preclinical trials. In my opinion, the beginning of the introduction should be rearranged.

Materials and Methods lacks data on the reagents and equipment used. In particular, starch is one of the main objects, what sample of it was used? From what raw materials? What are the characteristics?

SEM images of particles before and after enzymatic breakdown are good in some cases, but would enhance the work with more detailed photographs, as in reference 41, for example.

The main question is raised by Figure 5. Firstly, does it not duplicate the data in Figure 4? Secondly, why is the duration of treatment different for different enzyme preparations? And thirdly, taking into account the deviation, the advantage of amylase at the last point does not look significant.

Author Response

Dear Reviewer 2,

We greatly appreciate your insightful review and constructive comments on our manuscript. Your feedback has been instrumental in enhancing the clarity and depth of our work. We have made the recommended revisions and provided additional explanations to address your concerns. We hope that these improvements will meet your expectations and that our manuscript will be accepted for publication. Thank you once again for your valuable input and support.

--- *** ---

Reviewer 2

Manuscript polymers-3108684 provides new information about the importance of encapsulation in polyethylene glycol-coated starch microparticles of molecules with pharmacological interest to protect them from environments of the gastrointestinal tract. There are a number of shortcomings in the work.

It seems to me inappropriate to include "enzymes" in the title. There is insufficient data on both the most used enzymes (or enzyme preparations) and the mechanism of the studied enzymatic processes.

Response: Thank you for your observation. We respectfully disagree with the suggestion to remove "enzymes" from the title. The term "enzymes" is fundamental as our study investigates the enzymatic degradation of PEG-starch microparticles. The data presented, including the size reduction of microparticles over time due to enzyme action, directly highlights the enzymatic processes involved. Therefore, we believe that maintaining "enzymes" in the title accurately reflects the core focus and findings of our study.

In the introduction, the authors immediately tell about cancer, but the paper itself does not even contain data on preclinical trials. In my opinion, the beginning of the introduction should be rearranged.

Response: Thank you for this observation. We have revised the introduction to first contextualize the biomedical applications of polymeric microparticles. Although cancer is a primary focus of our research group, we have modified the introduction based on your suggestion to provide a broader context. Here is the added text with references:

Manuscript

Revised version

Introduction

Cancer is a complex multifactorial disease that…

Introduction

Polymeric microparticles have attracted great interest in recent years due to their versatile performance in various biomedical applications [1–5]. These microparticles can encapsulate a wide range of molecules, providing protection from degradation and facilitating their controlled release [6,7]. By encapsulating therapeutic agents, whether active ingredients or bioactive compounds, these microparticles improve solubility, stability and bioavailability with the potential to improve efficacy and selectivity in treatments. In recent years, much progress has been made in the development of polymeric microparticles designed to release their specific payload in response to stimuli, such as changes in pH [8]or enzymatic activity [9] which can be exploited for cancer therapy.

Cancer is a complex multifactorial disease that…

References added:

1.         Wang, M.; Wang, S.; Zhang, C.; Ma, M.; Yan, B.; Hu, X.; Shao, T.; Piao, Y.; Jin, L.; Gao, J. Microstructure Formation and Characterization of Long-Acting Injectable Microspheres: The Gateway to Fully Controlled Drug Release Pattern. Int J Nanomedicine 2024, 19, 1571–1595, doi:10.2147/IJN.S445269.

2.         Kupikowska-Stobba, B.; LewiÅ„ska, D. Polymer Microcapsules and Microbeads as Cell Carriers for in Vivo Biomedical Applications. Biomater Sci 2020, 8, 1536–1574, doi:10.1039/C9BM01337G.

3.         Bhujel, R.; Maharjan, R.; Kim, N.A.; Jeong, S.H. Practical Quality Attributes of Polymeric Microparticles with Current Understanding and Future Perspectives. J Drug Deliv Sci Technol 2021, 64, 102608, doi:10.1016/J.JDDST.2021.102608.

4.         Ma, G.; Yue, H. Advances in Uniform Polymer Microspheres and Microcapsules: Preparation and Biomedical Applications. Chin J Chem 2020, 38, 911–923, doi:10.1002/CJOC.202000135.

5.         Kupikowska-Stobba, B.; LewiÅ„ska, D. Polymer Microcapsules and Microbeads as Cell Carriers for in Vivo Biomedical Applications. Biomater Sci 2020, 8, 1536–1574, doi:10.1039/C9BM01337G.

6.         Wang, L.; Liu, Y.; Zhang, W.; Chen, X.; Yang, T.; Ma, G. Microspheres and Microcapsules for Protein Delivery: Strategies of Drug Activity Retention. Curr Pharm Des 2013, 19, 6340–6352, doi:10.2174/1381612811319350010.

7.         Zhang, Y.; Wei, W.; Lv, P.; Wang, L.; Ma, G. Preparation and Evaluation of Alginate-Chitosan Microspheres for Oral Delivery of Insulin. European Journal of Pharmaceutics and Biopharmaceutics 2011, 77, 11–19, doi:10.1016/J.EJPB.2010.09.016.

8.         Qiao, S.; Chen, W.; Zheng, X.; Ma, L. Preparation of PH-Sensitive Alginate-Based Hydrogel by Microfluidic Technology for Intestinal Targeting Drug Delivery. Int J Biol Macromol 2024, 254, doi:10.1016/J.IJBIOMAC.2023.127649.

9.         Thaarup, I.C.; Gummesson, C.; Bjarnsholt, T. Measuring Enzymatic Degradation of Degradable Starch Microspheres Using Confocal Laser Scanning Microscopy. Acta Biomater 2021, 131, 464–471, doi:10.1016/J.ACTBIO.2021.06.042.

Materials and Methods lacks data on the reagents and equipment used. In particular, starch is one of the main objects, what sample of it was used? From what raw materials? What are the characteristics?

Response: Thank you for your questions. Indeed, it is necessary to clarify the origin of the reagents, especially considering that starch can be derived from various sources. We have added details about the origin of the starch and other synthesis reagents in the Materials section. Here is the added text in the revised version:

Manuscript

Revised version

No Materials section

Materials

Starch from rice (CAS 9005-25-8; S7260), poly(ethylene glycol) (PEG, BioUltra 2,000, CAS 25322-68-3; 84797), folic acid (CAS 59-30-3, F7876) were purchased from Sigma-Aldrich, while acetone (Golden Bell, ACS), hydrochloric acid (J.T. Baker, ACS) and ammonium hydroxide (Fermont, ACS) were purchased separately. All reagents were used as received without any further purification.

SEM images of particles before and after enzymatic breakdown are good in some cases, but would enhance the work with more detailed photographs, as in reference 41, for example.

Response: Thank you for your suggestion. In this case, we focused on obtaining micrographs that provided a general idea of the morphology and allowed us to measure particle sizes using specialized software, rather than capturing highly detailed images. This approach was chosen to effectively demonstrate the overall structural changes and size variations before and after enzymatic breakdown, which are critical for our analysis. While we acknowledge the value of more detailed photographs as mentioned in reference 41, we believe that our chosen method adequately supports the objectives of our study by emphasizing the morphological and dimensional aspects essential for understanding the particles' behavior under enzymatic conditions.

The main question is raised by Figure 5. Firstly, does it not duplicate the data in Figure 4? Secondly, why is the duration of treatment different for different enzyme preparations? And thirdly, taking into account the deviation, the advantage of amylase at the last point does not look significant.

Response: Thank you for your concerns. We will address each of your questions individually.

Firstly, does it not duplicate the data in Figure 4?

Response: While the graph in Figure 5 is derived from the data in Figure 4, they serve different purposes. Figure 4 presents individual SEM micrographs and size histograms, allowing for detailed observation of particle morphology and size distribution at specific time points. However, these individual images do not clearly illustrate the changes over time. Figure 5 was created to provide a clear visualization of the size reduction trend over time. To clarify this distinction, we have added the following text to the manuscript:

Manuscript

Revised Version

Figure 5 examines

Figure 5 consolidates the time-dependent changes in microparticle size, derived from the individual data points presented in Figure 4 and it examines

Secondly, why is the duration of treatment different for different enzyme preparations?

Response: This is a good question. The durations were chosen based on the specific actions of the enzymes in simulated physiological conditions. For amylase, the time points of 5, 15, and 180 minutes were selected because amylase is primarily found in saliva, and saliva typically remains in the mouth for about 5 minutes before digestion continues. The 180-minute mark allows for comparison with simulated gastric conditions over an extended period. In the case of simulated gastric acid, the time points of 30, 60, and 180 minutes were chosen because these intervals are necessary for the natural digestion process. Although food typically spends 40 minutes to two hours in the stomach, we selected these intervals for our experiment to ensure thorough analysis. The reference for these durations can be found in Table 1 of the article (DOI: 10.1016/j.crfs.2021.04.004). We have added this explanation to the revised manuscript and the reference:

Manuscript

Revised Version

… the degradation was set at 30, 60 and 180 min, in the case of amylase pH 6.7, and the times were set at 5, 15 and 180 min. Approximately…

… the degradation was set at 30, 60 and 180 min, in the case of amylase pH 6.7, and the times were set at 5, 15 and 180 min. The different durations for enzyme treatments were chosen to reflect their specific roles in simulated physiological conditions, providing relevant time points for comprehensive analysis [54]. Approximately…

Reference added:

54.       Sensoy, I. A Review on the Food Digestion in the Digestive Tract and the Used in Vitro Models. Curr Res Food Sci 2021, 4, 308, doi:10.1016/J.CRFS.2021.04.004.

And thirdly, taking into account the deviation, the advantage of amylase at the last point does not look significant.

Response: You are correct. We have conducted a statistical analysis to address this point and clarify the observed data. The analysis results are now included in the manuscript to provide a more robust comparison and support our conclusions.

Manuscript

Revised Version

No analysis

Statistical Analysis

A statistical analysis using one-way Analysis of Variance (ANOVA) followed by Tukey's Honestly Significant Difference (HSD) post-hoc test was conducted. ANOVA was used to determine if there were significant differences in the average size of the microparticles over time for each enzyme treatment. Tukey's HSD test was applied post-hoc to identify specific time points with statistically significant differences in particle size.

In Results and Discussion section:

The size reduction of microparticles over time under the action of amylase and pepsin enzymes was observed. However, statistical analysis shows that these reductions are not significantly different across the enzyme treatments. Based on the data, an ANOVA test followed by Tukey's HSD test to assess the differences in microparticle size over time in the presence of amylase and pepsin enzymes were conducted. With a p-value greater than 0.05, the ANOVA test suggests that there is no statistically significant difference in the average size of the microparticles when comparing the groups treated with different enzymes. According to the Tukey's HSD test, there are no statistically significant differences between any pairs of groups at the 0.05 significance level. This indicates that while both enzymes contribute to the degradation of microparticles, their effects are not significantly distinct from each other or from the untreated control within the time frame studied.

Round 2

Reviewer 1 Report

Comments and Suggestions for Authors

The authors have implemented all the recommended alterations proposed by this reviwer. The quality of the manuscript has been raised by this and many incomplete sections are now more complete.

Author Response

Thank you for your thorough review and valuable suggestions. 

Reviewer 2 Report

Comments and Suggestions for Authors

Thanks to the authors, they did a decent job and certainly improved the manuscript. However, the methods do not specify the characteristics of the enzymes used (amylase and pepsin). Are these commercial samples or isolated by the authors? Enzymes are an important component of this work.

Author Response

Comment: Thanks to the authors, they did a decent job and certainly improved the manuscript. However, the methods do not specify the characteristics of the enzymes used (amylase and pepsin). Are these commercial samples or isolated by the authors? Enzymes are an important component of this work.
Response: Thank you again for your feedback. We apologize for the omission regarding the characteristics of the enzymes used in our study. The enzymes, alpha-amylase (E.C. 3.2.1.1.) and pepsin (E.C. 3.4.23.1.), are indeed important components of our work, and we acknowledge that their specific details should have been included. These enzymes were obtained as commercial food-grade products. Although we did not purify these enzymes further, their grade is suitable for the purpose of our study. We appreciate your suggestion and will include this information in the revised manuscript to ensure clarity and completeness.

We have added this information to the Materials section to provide a complete and transparent description of our experimental setup.

Materials:
Starch from rice (CAS 9005-25-8; S7260), poly(ethylene glycol) (PEG, BioUltra 2,000, CAS 25322-68-3; 84797), folic acid (CAS 59-30-3, F7876) were purchased from Sigma-Aldrich, while acetone (Golden Bell, ACS), hydrochloric acid (J.T. Baker, ACS) and ammonium hydroxide (Fermont, ACS) were purchased separately. All reagents were used as received without any further purification. The enzymes α-amylase (E.C. 3.2.1.1., 32,000-36,000 BAU/g ) and pepsin (E.C. 3.4.23.1., 1:10,000), both food grade, were obtained commercially.